# An Efficient and Mild Method for the Alkylation of p-Cresol with tert-Butyl Alcohol

Qi Wu [1], Dejin Zhang [1,2,*], Shu Sun [3], Chengcheng Liu [1] and Cong Wang [1]

1    School of Chemistry and Chemical Engineering, Suzhou University, Suzhou 234000, China
2    International Innovation Center for Forest Chemicals and Materials, Nanjing Forestry University, Nanjing 210037, China
3    School of Biology and Food Engineering, Suzhou University, Suzhou 234000, China
*    Correspondence: szxyzdj@163.com

**Abstract:** The synthesis 2-tert-butyl-4-methylphenol is of great significance because of its wide application in industry, and the development of a highly efficient catalyst is necessary for the alkylation of p-cresol and tert-butyl alcohol. Here, an efficient and mild method was established. Caprolactam was chosen as the hydrogen-bonding acceptor; p-toluenesulfonic acid was employed as the hydrogen-bonding donor, and a deep eutectic solvent (DES) was prepared to catalyze the alkylation reaction. The structure of the deep eutectic solvent catalyst was characterized by $^1$H NMR spectra, thermogravimetric analysis, and Fourier transform infrared spectra (FT-IR). In addition, response surface design based on the Box–Behnken method was employed to optimize the alkylation reaction process parameters, and the study of reaction kinetics was also carried out subsequently. The recycle performance of the catalyst was evaluated by recovery experiments, and a good result was obtained. By drawing comparisons with the literature reported, we provide a mild method for the synthesis of 2-tert-butyl-4-methylphenol.

**Keywords:** alkylation reaction; deep eutectic solvent; mild; response surface methodology





## 1. Introduction

The efficient synthesis of 2-tert-butyl-4-methylphenol (2-TBM) is an important reaction, because of its great application in the preparation of many kinds of fine chemicals, additives in the food industry, UV absorbers, and polymerization inhibitors [1–4]. Isobutylene, chlorhydrocarbon, tert-butyl methyl ether, and tert-butyl alcohol were employed as alkylating agents for the preparation of 2-TBM, and many synthesis methods were developed; however, the alkylation of p-cresol and tert-butyl alcohol to synthesize 2-TBM has been proved to be the most effective way because of the convenient operation in the reaction process. The catalyst is of great importance for the alkylation of p-cresol and tert-butyl alcohol to synthesize 2-TBM; an efficient catalyst can reduce the reaction energy barrier, and much milder reaction conditions can be used in the reaction system. According to the literature reported, both heterogeneous and homogeneous catalysts for the preparation of 2-TBM have been reported; however, these catalyst systems are the least preferred because they generate many problems, such as corrosion, environmental safety, and catalyst recyclability [5–8]. Hence, the ecofriendly and commercially viable catalyst systems are still in demand for alkylation reactions to synthesize 2-TBM.

Over the past few decades, an increasing interest has been focused on developing an ecofriendly catalyst system, and a deep eutectic solvent has attracted much attention because of its advantages, such as environment-friendly, easy to synthesize, high thermal stability, low vapor pressure, low volatility, and low cost [9–15]. As early as 2002, Abbott et al. found that the mixtures of quaternary ammonium salts with amides can form low melting point eutectics, and the unusual solvents were strongly influenced by hydrogen bonds [16]. After 20 years of rapid development, deep eutectic solvents have been wildly

applied in the extraction of biomass, absorption of acidic gas, organic synthesis, and other fields [17–20]. Particularly, different kinds of deep eutectic solvents have been used successfully as catalysts in esterification [21–24], oxidation [25–27], and alkylation reaction [28,29]. The wide application of deep eutectic solvents in chemical reactions is a green and efficient choice for catalytic reactions. However, as far as we know, there is still no literature reported on the application of deep eutectic solvents in the alkylation of p-cresol and tert-butyl alcohol to prepare 2-TBM. Here, the alkylation reaction catalyzed by the deep eutectic solvent was investigated, caprolactam (CAL) was chosen as the hydrogen-bonding acceptor, and p-toluenesulfonic acid (TsOH) was employed as the hydrogen-bonding donor. In addition, the response surface design based on the Box–Behnken method was employed to optimize the alkylation reaction process parameters, and the study of reaction kinetics was also carried out subsequently. A method for the alkylation of p-cresol and tert-butyl alcohol was developed, which may provide an efficient and mild way for the synthesis of 2-TBM in industry.

## 2. Results and Discussion

### 2.1. Structure Characterization of CAL-TsOH

2.1.1. [1]H NMR and [13]C NMR Analysis for CAL-TsOH

In order to characterize the catalyst, [1]H NMR and the characterization of caprolactam, p-toluenesulfonic acid, and CAL-TsOH were employed firstly in our experiments, and the results are shown in Figure 1. For the pure p-toluenesulfonic acid, the chemical shifts of the aromatic ring were 7.47 and 7.10 ppm; however, the chemical shifts of the aromatic ring moved to 7.59 and 7.25 ppm, which indicated the formation of hydrogen bonds in CAL-TsOH. In addition, the [13]C NMR characterization was performed in Figure 2, and the details are listed as follows for [13]C NMR (100 MHz, $D_2O$) δ (ppm): 181.6 (C=O); 145.0, 142.0, 132.2, and 128.0 (benzene ring); 41.8, 36.0, 29.0, 27.6, 26.2, and 23.1 (-$CH_3$, -$CH_2$).

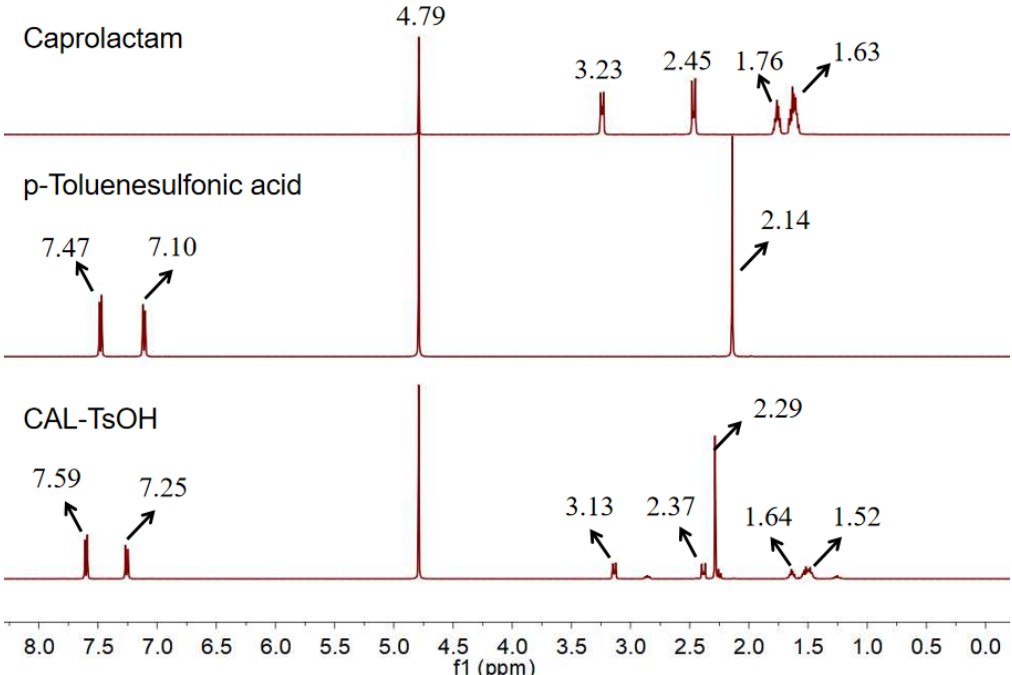

**Figure 1.** [1]H NMR of CAL-TsOH.

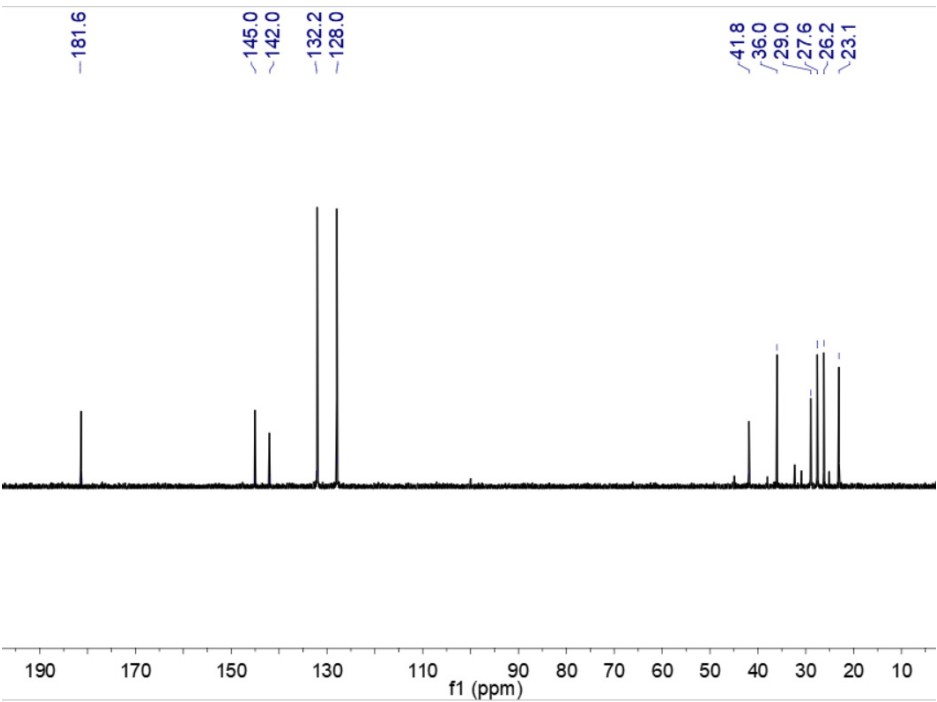

**Figure 2.** $^{13}$C NMR of CAL-TsOH.

### 2.1.2. FT-IR Spectra Analysis for CAL-TsOH

The FT-IR spectra for CAL-TsOH were determined, and the adsorption bands at 3413.3 and 3035.4 cm$^{-1}$ in Figure 3 were the stretching vibrations of O–H and N–H, and the hydrogen bonds in the deep eutectic solvent CAL-TsOH led to the broad adsorption peaks. The peaks appeared at 2946.7 and 2875.3 cm$^{-1}$ and were assigned to the stretching vibrations of CH$_3$ and CH$_2$, respectively. In addition, the characteristic peaks at 1677.7 and 1496.4 cm$^{-1}$ were assigned to the skeleton vibration of the benzene ring. In order to prove the structure of CAL-TsOH deeply, the HR-MS was also performed, the results are show in Figures S1–S4.

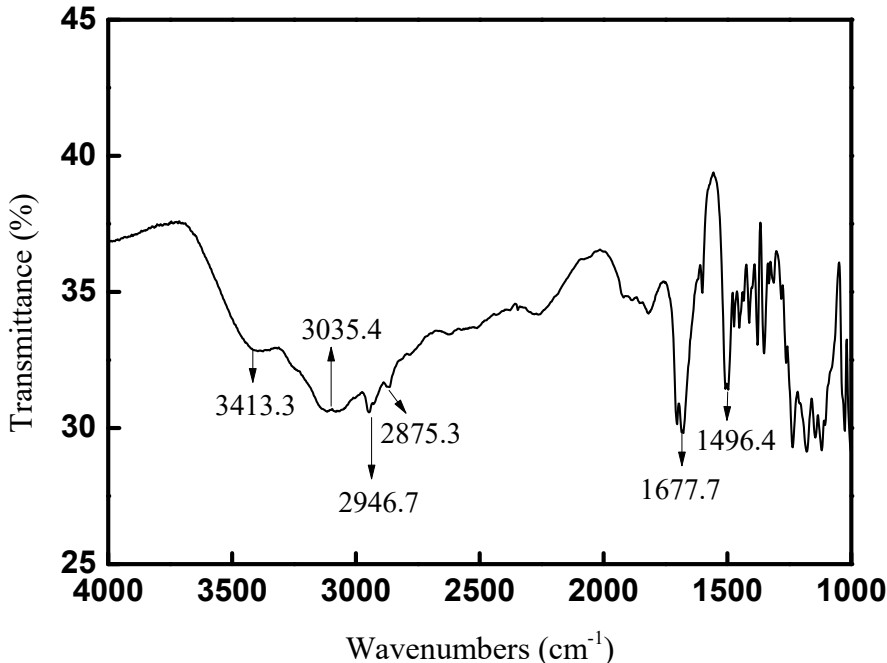

**Figure 3.** FT–IR spectra of CAL–TsOH.

2.1.3. Thermogravimetric Analysis for CAL-TsOH

Thermogravimetric analysis was also determined for CAL-TsOH, and the results are shown in Figure 4. A total of 5.7% of the mass loss of CAL-TsOH was obtained at 120 °C, which is due to the evaporation of little water in the deep eutectic solvent. After that, first a gentle, then a sharp decomposition process occurred in the TG curve of CAL-TsOH. The decomposition temperature was found as 180 °C, which indicated good thermostability of CAL-TsOH. In addition, DCS curve of CAL-TsOH was also determined in Figure S5, and the melting point was obtained.

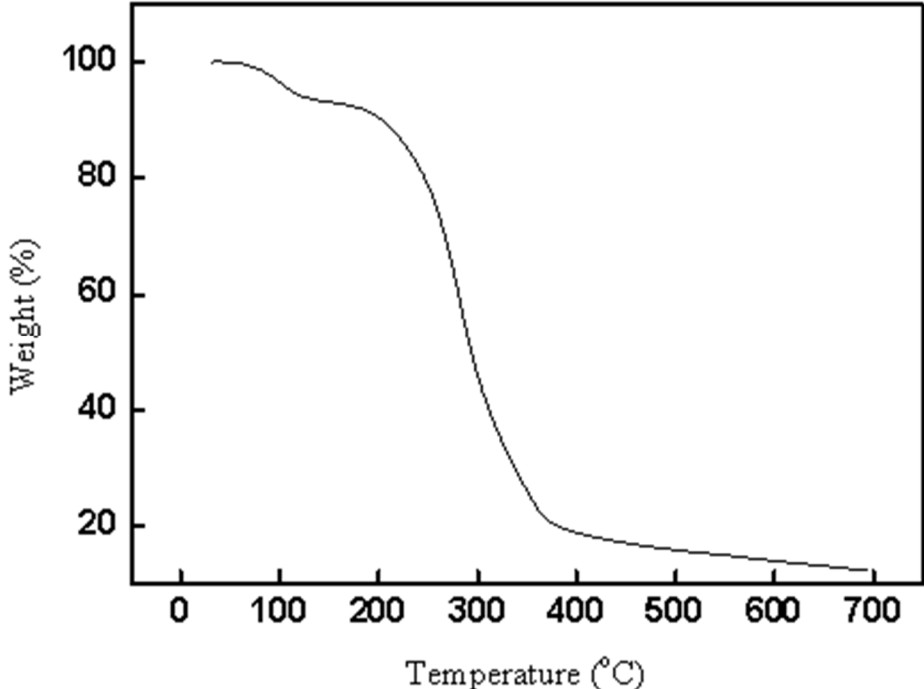

**Figure 4.** TG curve of CAL-TsOH.

*2.2. Optimization of the Conditions for the Synthesis of 2-TBM*

2.2.1. Effect of Mole Ratio (p-Cresol: tert-Butyl Alcohol) on the Conversion of tert-Butyl Alcohol

The effect of the mole ratio on the conversion of tert-butyl alcohol was explored firstly. A total of 5 mmol tert-butyl alcohol and 20 mol% (based on the amount of tert-butyl alcohol) CAL-TsOH were used, and the reaction was performed at room temperature for 10 h, and the results are shown in Figure 5. The results indicate the mole ratio of p-cresol and tert-butyl alcohol had a significant influence on the synthesis of 2-TBM. Only a 13% conversion was obtained when the mole ratio was set as 2; the conversion of tert-butyl alcohol increased with the increase in the mole ratio of p-cresol and tert-butyl alcohol; a 78% conversion was obtained when the mole ratio was set as 10; and a mole ratio of 10 was used in the following experiments.

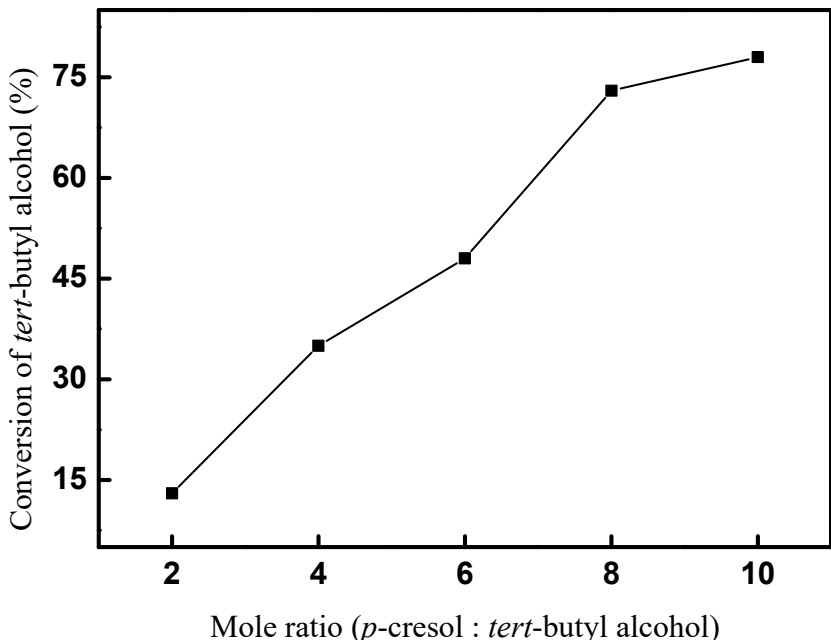

**Figure 5.** Effect of mole ratio on the conversion of tert-butyl alcohol.

### 2.2.2. Effect of Dosage of Catalyst on the Conversion of tert-Butyl Alcohol

The effect of the dosage of CAL-TsOH on the synthesis of 2-TBM was investigated, and 5 mmol tert-butyl alcohol and 50 mmol p-cresol were used in the reaction system. The alkylation reaction was performed at room temperature for 10 h, and the results are shown in Figure 6. The results indicate that the conversion of tert-butyl alcohol was significantly affected by the dosage of the catalyst; only a 24% conversion was obtained when the dosage of the catalyst was 5 mol% (based on the amount of tert-butyl alcohol), and the conversion of tert-butyl alcohol increased with the increase in the dosage of the catalyst; there was no obvious increase in the conversion in the synthesis reaction of 2-TBM after the dosage of the catalyst increased to 20 mol%. In consideration of the economic performance, 20 mol% of CAL-TsOH was used in our following experiments.

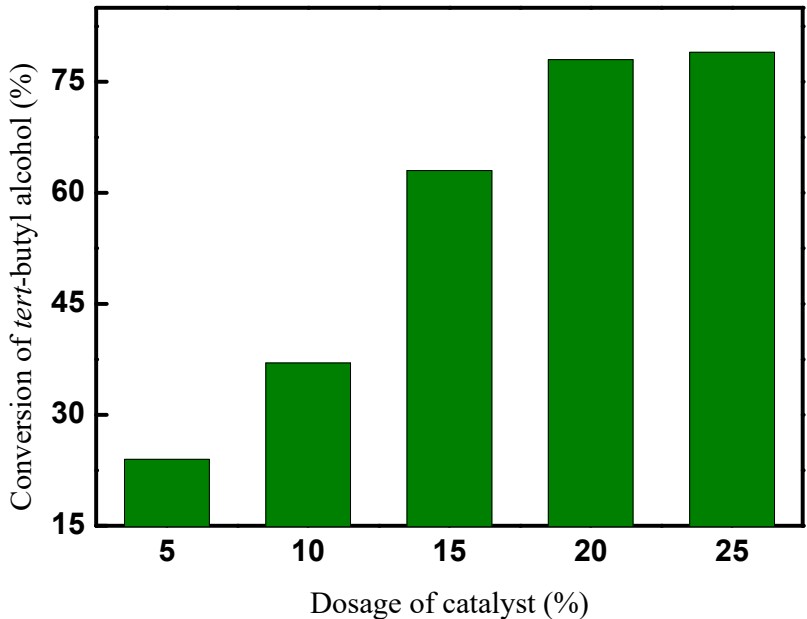

**Figure 6.** Effect of dosage of catalyst on conversion of tert-butyl alcohol.

### 2.2.3. Effect of Reaction Time on the Conversion of tert-Butyl Alcohol

A total of 5 mmol tert-butyl alcohol, 50 mmol p-cresol, and 20 mol% of catalyst were used in the reaction system. The effect of the reaction time on the synthesis of 2-TBM at room temperature was explored in our experiments, and the results are shown in Figure 7. The results imply that the synthesis of 2-TBM was significantly affected by the reaction time. Only a 16% conversion was obtained after reacting for 1 h, and the conversion of tert-butyl alcohol increased quickly in the next 5 h. A 53% conversion was obtained after 6 h for the reaction time, and then the conversion of tert-butyl alcohol increased slowly. However, there was still a little increase when the reaction time was 12 h; 10 h of reaction time was more suitable, and a 78% conversion of tert-butyl alcohol was obtained.

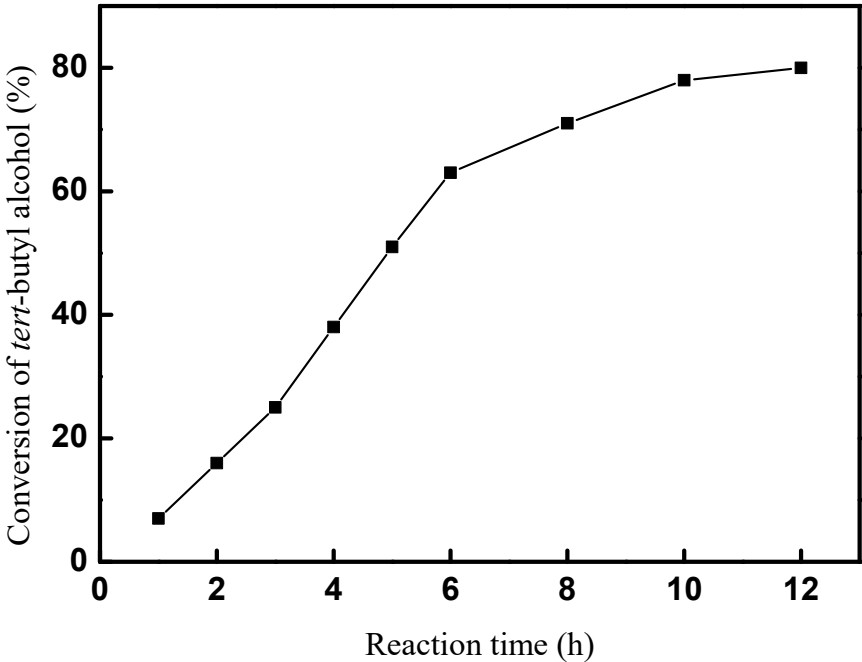

**Figure 7.** Effect of reaction time on conversion of tert-butyl alcohol.

### 2.3. Response Surface Methodology for the Alkylation of p-Cresol and tert-Butyl Alcohol

In this study, the three main variables were the mole ratio (A), dosage of catalyst (B), and reaction time (C), and the response factor of the conversion of tert-butyl alcohol was optimized by response surface methodology. According to the results of the single-factor experiments, the three levels of the mole ratio were set as 8, 10, and 12; the three levels of the dosage of the catalyst were set as 15, 20, and 25, and then the three levels of the reaction time were set as 8, 10, and 12. Seventeen groups of experiments were designed by response surface methodology. The experimental design was based on the Box–Behnken design method, and the results are shown in Table 1. The established model for the conversion of tert-butyl alcohol is listed as follows:

$$Conversion(\%) = 78 + 2.25A + 5.88B + 2.38C + 0.25AB + 0.75AC \\ + 3.00BC - 3.50A^2 - 5.25B^2 - 1.75C^2 \tag{1}$$

**Table 1.** Experimental design and results of response surface methodology.

| Entry | Mole Ratio | Dosage of Catalyst/mol% | Reaction Time/h | Conversion/% |
|---|---|---|---|---|
| 1 | 10 | 20 | 10 | 78 |
| 2 | 10 | 15 | 8 | 68 |
| 3 | 10 | 25 | 12 | 80 |
| 4 | 12 | 25 | 10 | 79 |
| 5 | 10 | 25 | 8 | 71 |
| 6 | 8 | 15 | 10 | 60 |
| 7 | 10 | 20 | 10 | 78 |
| 8 | 12 | 20 | 12 | 79 |
| 9 | 8 | 20 | 8 | 68 |
| 10 | 10 | 20 | 10 | 78 |
| 11 | 8 | 25 | 10 | 74 |
| 12 | 8 | 20 | 12 | 73 |
| 13 | 12 | 15 | 10 | 64 |
| 14 | 10 | 20 | 10 | 78 |
| 15 | 10 | 15 | 12 | 65 |
| 16 | 12 | 20 | 8 | 71 |
| 17 | 10 | 20 | 10 | 78 |

To investigate the goodness of the fit for the model, analysis of variance (ANOVA) was used in our research. The results for the analysis of variance are illustrated in Table 2. P-values of 0.0003 and 0.1346 for the model and lack of fit were obtained, respectively, which indicate the model in equation (1) was highly significant, and the established model fitted well in the current research [30].

**Table 2.** Analysis of variance.

| Source | Sum of Squares | df | Mean Square | F-Value | *p*-Value | |
|---|---|---|---|---|---|---|
| Model | 597.69 | 9 | 66.41 | 21.88 | 0.0003 | significant |
| A—mole ratio | 40.5 | 1 | 40.5 | 13.34 | 0.0082 | |
| B—dosage of catalyst | 276.13 | 1 | 276.13 | 90.96 | <0.0001 | |
| C—reaction time | 45.13 | 1 | 45.13 | 14.86 | 0.0062 | |
| AB | 0.25 | 1 | 0.25 | 0.082 | 0.7824 | |
| AC | 2.25 | 1 | 2.25 | 0.74 | 0.4178 | |
| BC | 36.00 | 1 | 36.00 | 11.86 | 0.0108 | |
| $A^2$ | 51.58 | 1 | 51.58 | 16.99 | 0.0044 | |
| $B^2$ | 116.05 | 1 | 116.05 | 38.23 | 0.0005 | |
| $C^2$ | 12.89 | 1 | 12.89 | 4.25 | 0.0783 | |
| Residual | 21.25 | 7 | 3.04 | | | |
| Lack of fit | 21.25 | 3 | 7.08 | 7.43 | 0.1346 | not significant |
| Pure error | 0.00 | 4 | 0.00 | | | |

In order to investigate the interaction influence of mole ratio (A), dosage of catalyst (B), and reaction time (C), 3D response surfaces were employed in Figure 8. The obvious interaction influence among mole ratio (A), dosage of catalyst (B), and reaction time (C) was found in our research because of the oval shapes of the contour lines. Subsequently, the optimization process was performed by Design Expert; an 83% conversion of tert-butyl alcohol was obtained with a 10.97 mole ratio, 24.17 mol% dosage of the catalyst, and 12.00 h of reaction time. To verify the optimization results by response surface methodology, an experiment was performed with a mole ratio of 11, 25 mol% dosage of the catalyst, and 12.00 h of reaction time. An 85% conversion of tert-butyl alcohol was obtained, and the results indicate the reliability of the optimization results by response surface methodology.

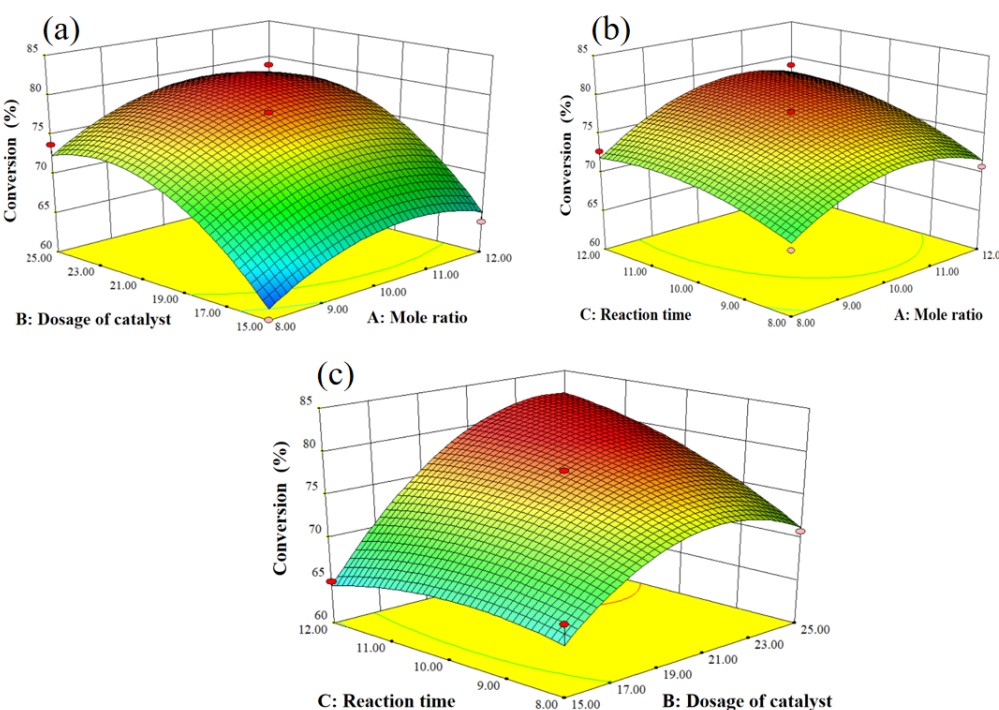

**Figure 8.** 3D response surface curves for the synthesis of 2-TBM: (**a**) dosage of catalyst and mole ratio, (**b**) reaction time and mole ratio, and (**c**) reaction time and dosage of catalyst.

### 2.4. Comparison of CAL-TsOH with Other Catalysts

To thoroughly compare the potential of different catalysts for the synthesis of 2-TBM, the reaction temperature and conversion reported in the literature are summarized in Table 3. Both heterogeneous and homogeneous catalysts for the preparation of 2-TBM have been reported. For example, Elavarasanit et al. reported the alkylation of p-cresol and tert-butyl alcohol catalyzed by ionic liquid. Response surface methodology was employed to optimize the process parameters, and an 89.4% conversion was obtained with a reaction temperature of 348 K [31]. Some other homogeneous catalysts were also used in the synthesis of 2-TBM. Density functional theory was used to develop a molecular level understanding of the alkylation of p-cresol and tert-butyl alcohol catalyzed by acidic ionic liquids; an 86% conversion was obtained under a reaction temperature of 343 K [32]. The $SO_3H$-functionalized ionic liquids were synthesized, and the catalytic performance for the alkylation of p-cresol and tert-butyl alcohol was evaluated, and a 79% conversion and 92% selectivity were obtained [33]. Kishore et al. reported some novel Brønsted acidic ionic liquids, N-methyl imidazole, pyridine, triethylamine, and 1,4-butanesultone, were used as the source chemicals, and an 80% conversion was obtained under a reaction temperature of 70 °C [34]. In addition, the alkylation of p-cresol and tert-butyl alcohol to 2-TBC catalyzed by multiple-$SO_3H$ functional ionic liquid was reported by Bao et al., and an 85.3% conversion and 95.2% selectivity were obtained [35]. Heterogeneous catalyst were also reported widely in the literature; however, a much higher reaction temperature was needed. $WOx/ZrO_2$ was prepared by the wet impregnation method; zirconium oxyhydroxide, and ammonium metatungstate were used; the catalyst 15% $WO_3/ZrO_2$ calcined at 800 °C was found to be the most active in the reaction; a 69.8% conversion and 92.4% selectivity of 2-tert-butyl-p-cresol were obtained under the optimized reaction conditions of 130 °C, with a tert-butanol/p-cresol molar ratio of 3 and a flow rate of 10 mL h$^{-1}$ [36]. Devassy et al. reported a 12-tungstophosphoric acid supported on zirconia ($TPA/ZrO_2$) under flow conditions; the effects of the molar ratio of p-cresol and tert-butyl alcohol, reaction temperature, and space velocity on the conversion of p-cresol and product selectivities were optimized, and a 61% conversion of p-cresol and an 81.4 selectivity of 2-tert-butyl-p-cresol were obtained under a reaction temperature of 403 K [37]. In addition, titania modified

with a 12-tungstophosphoric acid (TPA/TiO$_2$) catalyst was prepared by Kumbar et al., and surface area, XRD, $^{31}$P MAS NMR, XPS, NH$_3$-TPD, and FTIR pyridine adsorption were employed to characterize these catalysts, and an 82% conversion of p-cresol and an 89.5 selectivity of 2-tert-butyl-p-cresol were obtained under the optimized reaction conditions [38]. Long-chain double SO$_3$H-functionalized Brønsted acidic ionic liquids were synthesized, and the catalytic performances for the alkylation of p-cresol were investigated; an 89.4% conversion of the phenol, a 73.7% selectivity of 2,4-tert-butyl-phenol, a 93.2% conversion of p-cresol, and an 89.2% selectivity of 2-tert-butyl-p-cresol were obtained by Li et al. [39]. Additionally, the alkylation of p-cresol with MTBE to synthesize 2-tert-butyl-p-cresol was investigated, and a mesoporous and strong acid catalyst UDCaT-1 was prepared by Yadav et al. The ratio of p-cresol and MTBE was 1:1; the reaction temperature was set as 100 °C, and a 45% conversion was obtained [40].

**Table 3.** Comparison of catalytic performance for alkylation reaction.

| Entry | Catalyst | Temperature (K) | Conversion (%) | Refs |
|---|---|---|---|---|
| 1 | CAL-TsOH | 298 | 78 | This work |
| 2 | N-(1,4-sulfonic acid) butyl triethylammonium hydrogen sulfate | 348 | 89.4 | [31] |
| 3 | IL-CF$_3$SO$_3$ | 343 | 86.2 | [32] |
| 4 | SO$_3$H ionic liquids | 343 | 79 | [33] |
| 5 | SO$_3$H-functionalized Brønsted acidic ionic liquid | 343 | 80 | [34] |
| 6 | multiple-SO$_3$H ionic liquid | 343 | 85.3 | [35] |
| 7 | WOx/ZrO$_2$ | 403 | 69.8 | [36] |
| 8 | TPA/ZrO$_2$ | 403 | 61 | [37] |
| 9 | TPA/TiO$_2$ | 403 | 82 | [38] |
| 10 | BAIL-1 | 343 | 93.2 | [39] |
| 11 | UDCaT-1 | 373 | 45 | [40] |

The reaction temperature in our experiments is clearly much milder than those catalyzed by other catalysts reported in the literature, and a satisfactory conversion of tert-butyl alcohol was obtained.

*2.5. Reaction Kinetics for the Alkylation of p-Cresol and tert-Butyl Alcohol*

In consideration of the application for the synthesis of 2-TBM catalyzed by CAL-TsOH in industry, the investigation of reaction kinetics for the alkylation of p-cresol and tert-butyl alcohol is essential, and the reaction equation can be expressed in Scheme 1.

**Scheme 1.** The reaction equation for the synthesis of 2-TBM.

However, no butylated hydroxytoluene (BHT) was found in our reaction system; 2-tert-Butyl-4-methylphenol (2-TBM) was the only product. Hence, the kinetic studies were carried out in standard reaction conditions based on the follow assumptions: (a) the intermediate products can be neglected, and (b) the reverse reaction can be ignored [41,42]. The equation of the reaction rate can be expressed as follows:

$$r = kC_A C_B \tag{2}$$

The excessive p-cresol (p-cresol: tert-butyl alcohol = 10) was used in our experiments; thus, the concentration of p-cresol has little influence on the alkylation reaction rate. Consequently, the kinetic rate equation for the synthesis of 2-TBM at room temperature catalyzed by CAL-TsOH can be expressed as follows:

$$r = -\frac{dC_B}{dt} = kC_B \tag{3}$$

$$C_B = C_{B0}(1 - y) \tag{4}$$

where $k$ is the alkylation reaction rate constant, $h^{-1}$; $C_{B0}$ and $C_B$ are the concentrations of tert-butyl alcohol at the beginning and end of the synthesis of 2-TBM, $mol \cdot L^{-1}$; $y$ (%) is the conversion of tert-butyl alcohol determined by GC; t is the alkylation reaction time, h.

The Equation (3) can be transformed to the linearization form as:

$$-\ln(1 - y) = kt \tag{5}$$

Therefore, the reaction rate constant k can be calculated by plot of $-\ln(1 - y)$ versus t. The fitted results of the kinetic study at room temperature are shown in Figure 9; the linear correlation of $R_2$ was 0.975, which indicated the goodness of fit for the alkylation of tert-butyl alcohol and p-cresol; 0.18537 $h^{-1}$ of k was obtained, and the kinetic equation for the alkylation of tert-butyl alcohol and p-cresol under room temperature was obtained as $-\ln(1 - y) = 0.18537t - 0.1708$. The kinetic study played an important role in the industrialization application for the synthesis of 2-TBM under room temperature.

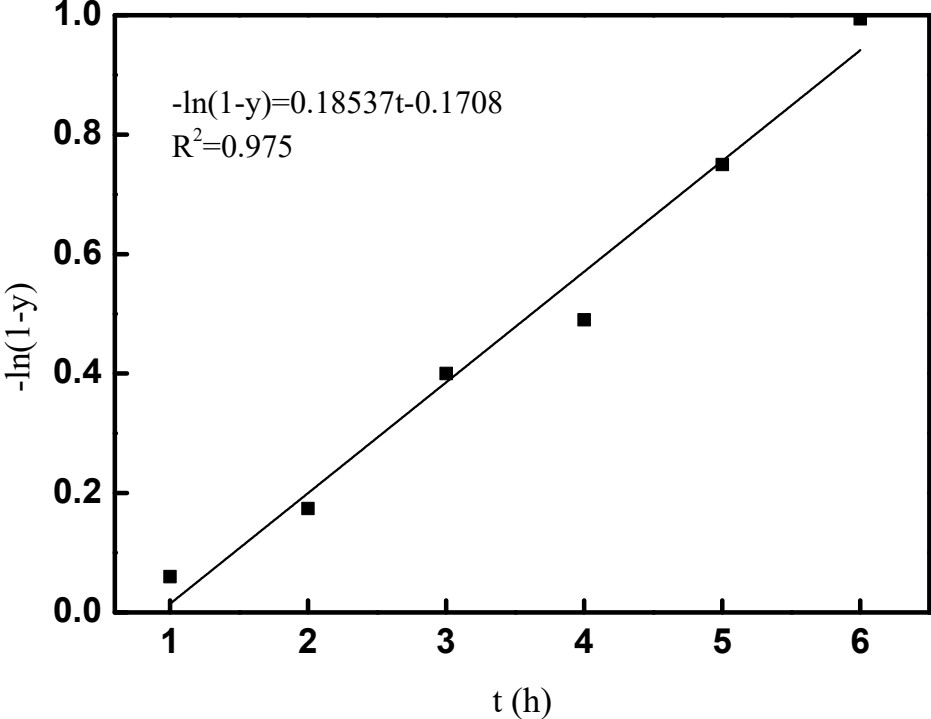

**Figure 9.** Fitting curve of $-\ln(1 - y)$ and t.

## 2.6. Catalyst Recovery for the CAL-TsOH

The recycle performance of CAL-TsOH was investigated by a recovery experiment. Firstly, tert-butyl alcohol, p-cresol, and CAL-TsOH were mixed in a 25 mL reaction tube. A homogeneous solution was obtained, and then the alkylation reaction was performed in the standard reaction condition. After the reaction, the catalyst in the reaction tube was extracted by ethyl acetate, and CAL-TsOH was obtained in the lower layer. The recycled

CAL-TsOH was washed by ethyl acetate several times and dried for 24 h at 70 °C. The results of the catalyst recovery experiment are displayed in Figure 10. The conversion of tert-butyl alcohol decreased gradually after 5 catalytic cycles, and a 68% conversion was obtained, which may due to the loss of CAL-TsOH during the extraction process by ethyl acetate.

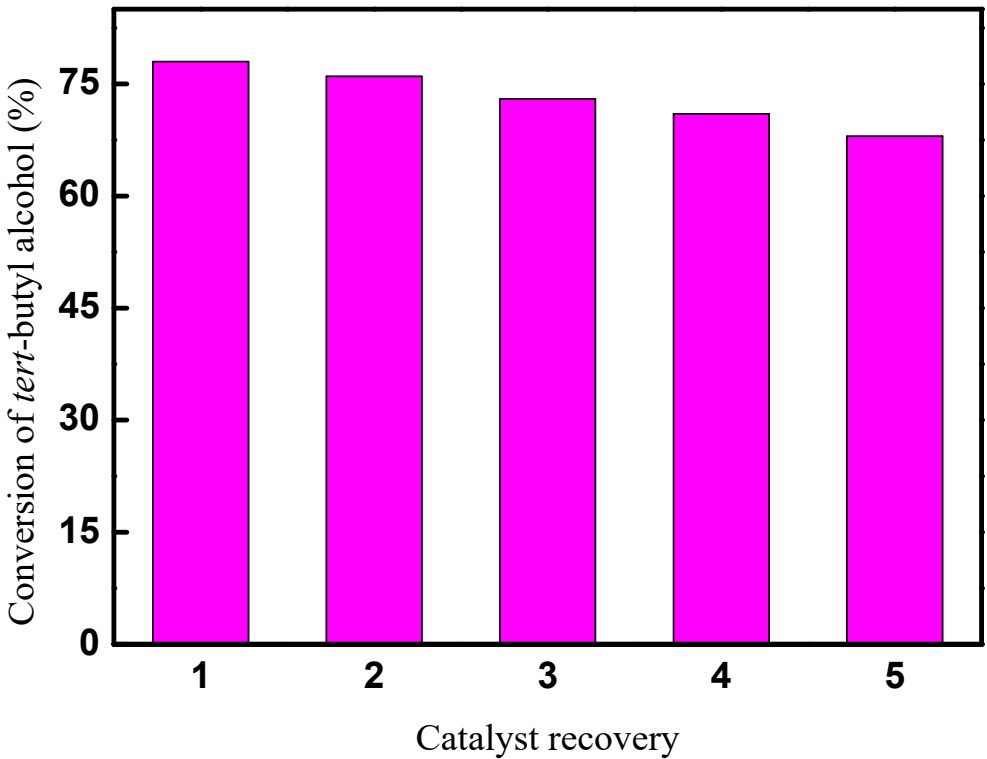

**Figure 10.** Results of recovery experiment for CAL-TsOH.

### 3. Materials and Methods

*3.1. Materials and Chemicals*

Caprolactam and tert-butyl alcohol were obtained from Sinopharm Chemical Reagent Co., Ltd. (Suzhou, China); p-toluenesulfonic acid (TsOH) was purchased from Aladdin Reagent Co., Ltd. (Shanghai, China); p-cresol was purchased from Energy Chemical Reagent Co., Ltd. (Hefei, China); ethyl acetate was purchased from Nanjing Chemical Reagent Co., Ltd. (Nanjing, China); All of the reagents in this research were used without further purification.

*3.2. Preparation of CAL-TsOH*

Caprolactam was chosen as hydrogen-bonding acceptor; p-toluenesulfonic acid was employed as hydrogen-bonding donor. A total of 3.44 g p-toluenesulfonic acid and 2.26 g caprolactam were mixed and stirred at 80 °C for 24 h in a 25 mL round-bottomed flask. Then, a homogeneous and clear liquid of CAL-TsOH was obtained.

*3.3. Characterization of CAL-TsOH*

Bruker Avance 400 spectrometer was employed to record $^1$H NMR of CAL-TsOH; $D_2O$ was used as solvent. FT-IR spectra of CAL-TsOH were obtained by Nicolet iS50 spectrometer. In order to investigate the thermostability of the catalyst, thermogravimetric analysis (TGA) was performed under $N_2$ atmosphere, and the heat rate was set as 10 °C/min from 30 to 700 °C.



*3.4. Procedure for the Synthesis of 2-TBM*

A total of 50 mmol p-cresol and 5 mmol tert-butyl alcohol were mixed in a 25 mL reaction tube, and 20 mol% (based on the amount of tert-butyl alcohol) of CAL-TsOH was used. The reaction tube was put into a parallel apparatus, which was equipped with condensation unit subsequently. The reaction tube was stirred for 10 h at room temperature subsequently. Then, the CAL-TsOH catalyst in reaction tube was extracted by ethyl acetate, and gas chromatography was used to determine the conversion of tert-butyl alcohol.

*3.5. Single-Factor Experiments*

3.5.1. Effect of Mole Ratio on the Conversion of tert-Butyl Alcohol

The effect of mole ratio on the conversion of tert-butyl alcohol was performed firstly. A total of 5 mmol tert-butyl alcohol and 20 mol% CAL-TsOH were used in our reaction system. The reaction was performed at room temperature for 10 h, and the mole ratio of p-cresol to tert-butyl alcohol was set as 2, 4, 6, 8, and 10. The conversion of tert-butyl alcohol was determined by GC to investigate the effect of mole ratio.

3.5.2. Effect of Dosage of Catalyst on the Conversion of tert-Butyl Alcohol

The effect of dosage of catalyst on the conversion of tert-butyl alcohol was investigated. A total of 5 mmol tert-butyl alcohol and 50 mmol p-cresol were used. The dosage of catalyst was set as 5, 10, 15, 20, and 25 mol% CAL-TsOH (based on tert-butyl alcohol) in our reaction system, and the reaction was performed at room temperature for 10 h. After the reaction, the conversion of tert-butyl alcohol was determined by GC to investigate the effect of dosage of catalyst.

3.5.3. Effect of Reaction Time on the Conversion of tert-Butyl Alcohol

The effect of reaction time on the conversion of tert-butyl alcohol was investigated. A total of 5 mmol tert-butyl alcohol, 50 mmol p-cresol and 20 mol% of catalyst were used, and the effect of reaction time on the synthesis of 2-TBM at room temperature was investigated in our reaction system. After the reaction, the conversion of tert-butyl alcohol was determined by GC to investigate the effect of reaction time.

*3.6. Experimental Design of Response Surface Methodology*

In order to investigate the interaction influence of mole ratio (A), dosage of catalyst (B), and reaction time (C), response surface methodology was used in our experiment. According to the results of single-factor experiments, three-factor and three-level experiments were designed based on Box–Behnken method. The conversion of tert-butyl alcohol was set as response value, and the synthesis of 2-TBM process was optimized by response surface methodology.

*3.7. Determination of the Conversion of tert-Butyl Alcohol*

Shimadzu GC 2014C was employed to determine the conversion of tert-butyl alcohol, and FID detector and RTX-5 capillary column were equipped with the GC. The temperatures of FID detector and gasification chamber were set as 300 °C; the temperature of the column was 40 °C initially and held for 2 min, and then the column was programmed to heat up until 280 °C at a heat rate of 30 °C/min and held for 3 min. In addition, the conversion of tert-butyl alcohol was calculated according to the determination results by GC. The calculated equation can be expressed as follows:

$$Conversion = \frac{n_{B0} - n_B}{n_{B0}} \times 100\% \tag{6}$$

$n_{B0}$ and $n_B$ are the mole numbers of tert-butyl alcohol at the beginning and end of the synthesis of 2-TBM.

## 4. Conclusions

An efficient and mild method was established; caprolactam was chosen as the hydrogen-bonding acceptor; p-toluenesulfonic acid was employed as the hydrogen-bonding donor, and deep eutectic solvent was prepared to catalyze the alkylation reaction of p-cresol and tert-butyl alcohol. $^1$H NMR spectra, thermogravimetric analysis, and FT-IR spectra were used to characterize CAL-TsOH. Response surface methodology was employed to optimize the conditions for the preparation of 2-TBM, an 83% conversion of tert-butyl alcohol was obtained with a 10.97 mole ratio, 24.17 mol% of dosage of catalyst, and 12.00 h of reaction time. In addition, the reaction kinetics and a reaction rate constant k of 0.18537 h$^{-1}$ were obtained. The recovery experiment was also used to evaluate the recycle performance; the conversion decreased gradually after five catalytic cycles, and a 68% conversion was obtained. Compared with other reports, the reaction temperature in our experiments is clearly much milder than those catalyzed by other catalysts reported, and a satisfactory conversion of tert-butyl alcohol was obtained. The method provides a mild way for the synthesis of 2-TBM in industry.

**Supplementary Materials:** The following supporting information can be downloaded at: https://www.mdpi.com/article/10.3390/catal13061002/s1, Figure S1. HR-MS of CAL-TsOH (m/z = 50–500); Figure S2. HR-MS of CAL-TsOH (m/z = 50–180); Figure S3. HR-MS of CAL-TsOH (m/z = 180–280); Figure S4. HR-MS of CAL-TsOH (m/z = 260–500); Figure S5. DSC curve of CAL-TsOH.

**Author Contributions:** Experimental investigation, Q.W.; writing original draft, S.S.; review and supervision, D.Z.; kinetics study, C.L. supervision, C.W.; All authors have read and agreed to the published version of the manuscript.

**Funding:** This work was funded by the Natural Science Research Project of the Education Department of Anhui Province (2022AH051364), National Innovation and Entrepreneurship Training Program for college students (202210379008), Suzhou Science and Technology Planning Project (2021033), and doctoral research start-up fund of Suzhou University (2021BSK054), Science and technology development fund project of Suzhou University (2021fzjj08).

**Data Availability Statement:** Available upon request.

**Acknowledgments:** This work was supported by the Suzhou University school-level scientific research platform, and the head of the platform, Hongwei Shi, provided some important suggestion for the work.

**Conflicts of Interest:** The authors declare no conflict of interest.

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
