# Peer review of "An Efficient and Mild Method for the Alkylation of p-Cresol with tert-Butyl Alcohol"

_catalysts, doi:10.3390/catal13061002_

Round 1
Reviewer 1 Report
The authors have demonstrated the use of the deep eutectic solvent CAL -TsOH for the preparation of 2-tert-butyl-4-methylphenol. A full characterisation of the catalyst was carried out and the optimisation process and reaction kinetics were very well studied.
Unfortunately, the English style is poor and it is difficult to understand some points correctly.
1) The introduction is confusing, with several repetitions. Cat-TsOH needs to be introduced and a list of previous applications needs to be cited appropriately.
2) The comparison of CAL -TsOH with other catalysts also needs to be done in terms of energy consumption and green metrics.
3) The conclusion is also quite confusing due to poor use of English.
See below
Author Response
The authors have demonstrated the use of the deep eutectic solvent CAL -TsOH for the preparation of 2-tert-butyl-4-methylphenol. A full characterisation of the catalyst was carried out and the optimisation process and reaction kinetics were very well studied.
Unfortunately, the English style is poor and it is difficult to understand some points correctly.
Response : Dear reviewer, thank you very much for taking your time to review our manuscript submitted to Catalysts. We also want to express our appreciation for your valuable suggestions. The followings are our response to your comments.
Point 1: The introduction is confusing, with several repetitions. Cat-TsOH needs to be introduced and a list of previous applications needs to be cited appropriately.
Response 1: Thanks for your suggestion. We have checked the introduction carefully, and some unsuitable description has been reduced, and the applications of the deep eutectic solvent have been added in our manuscript.
Point 2: The comparison of CAL -TsOH with other catalysts also needs to be done in terms of energy consumption and green metrics.
Response 2: Thanks. The comparison of CAL -TsOH with other catalysts has been performed in 2.4. It can be clearly seen that the reaction temperature in our experiments is much milder than those catalyzed by other catalysts reported in the literature, and a satisfactory conversion of tert-butyl alcohol was obtained.
Point 3: The conclusion is also quite confusing due to poor use of English.
Response 3: Thanks for your suggestion. We are sorry for the mistakes, the description has been revised especially in the last sentence of the conclusion.

Reviewer 2 Report
In this work, Wu, Zhang et al. reported an efficient method for the alkylation of p-cresol
with tert-butyl alcohol. It's interesting that caprolactam was chosen as hydrogen-bonding acceptor, p-toluenesulfonic acid was employed as hydrogen-bonding donor, and deep eutectic solvent (DES) was prepared to catalyze the alkylation reaction. More important, response surface methodology was employed to optimize the conditions for the preparation of 2-TBM, 83% conversion of tert-butyl alcohol was obtained with 10.97 of mole ratio, 24.17 mol% of dosage of catalyst and 12.00 hours of reaction time. I would like to recommend this work to Catalysts.
Author Response
In this work, Wu, Zhang et al. reported an efficient method for the alkylation of p-cresol with tert-butyl alcohol. It's interesting that caprolactam was chosen as hydrogen-bonding acceptor, p-toluenesulfonic acid was employed as hydrogen-bonding donor, and deep eutectic solvent (DES) was prepared to catalyze the alkylation reaction. More important, response surface methodology was employed to optimize the conditions for the preparation of 2-TBM, 83% conversion of tert-butyl alcohol was obtained with 10.97 of mole ratio, 24.17 mol% of dosage of catalyst and 12.00 hours of reaction time. I would like to recommend this work to Catalysts.
Response : Dear reviewer, thank you very much for taking your time to review our manuscript submitted to Catalysts. We also want to express our appreciation for your valuable suggestions.
Reviewer 3 Report
The present submission “An Efficient and Mild Method for the Alkylation of p-Cresol with tert-Butyl Alcohol” described the synthesis of an organic acidic compound as catalyst for dehydration reaction leading to a fine chemical. In general, the use of organic catalyst would avoid the pollution of product derived from metal ions, and meanwhile the target molecule (2-TBM) showed wide applications in many areas. Furthermore, the total manuscript was well organized and written. Obviously, this work would contribute to the large-scale production of 2-TBM. However, there were several drawbacks deserved attentions and corresponding revisions. After improvements, I would like to recommend this work for publication in Catalysts.
At first, in Sect. Introduction, the catalytic synthesis of 2-tert-butyl-4-methylphenol (2-TBM) was not well reviewed. For example, the classical synthesis of 2-TBM should be carefully illustrated, including the known synthetic route, starting materials, advantages and disadvantages. In particular, how about the utilization of catalysts. These issues should be carefully reviewed and discussed, and the present texts seemed too simple and lack of persuasiveness. Kindly suggest that the authors should let readers to believe the importance of the present work.
Secondly, as for the characterizations of organic compounds including catalysts, in addition to 1H NMR, FT-IR and TGA, 13C NMR, HR-MS, elemental analysis, melting point should be added. I have to clearly point out that only 1H NMR, FT-IR and TGA could not illustrate the structures of organic compounds including catalysts. This is very important, please also discuss with other experts on this point.
Next, very necessary, the authors will have to study the by-products of this kind of transformation, through GC-MS or HPLC-MS. Are there any by-products like esters or carbonyl compounds? Some key experimental evidences should be provided.
At last, the authors should summarize catalytic mechanism, which meant a lot to understanding the role of organo-catalyst in transformation.
If the above issues could be revised and improved, I would like to recommend this work for publication in Catalysts.
Yes, quality of English language is OK, and can be polished at the end of revision.
Author Response
The present submission “An Efficient and Mild Method for the Alkylation of p-Cresol with tert-Butyl Alcohol” described the synthesis of an organic acidic compound as catalyst for dehydration reaction leading to a fine chemical. In general, the use of organic catalyst would avoid the pollution of product derived from metal ions, and meanwhile the target molecule (2-TBM) showed wide applications in many areas. Furthermore, the total manuscript was well organized and written. Obviously, this work would contribute to the large-scale production of 2-TBM. However, there were several drawbacks deserved attentions and corresponding revisions. After improvements, I would like to recommend this work for publication in Catalysts.
Response : Dear reviewer, thank you very much for taking your time to review our manuscript submitted to Catalysts. We also want to express our appreciation for your valuable suggestions. The followings are our response to your comments.
Point 1: At first, in Sect. Introduction, the catalytic synthesis of 2-tert-butyl-4-methylphenol (2-TBM) was not well reviewed. For example, the classical synthesis of 2-TBM should be carefully illustrated, including the known synthetic route, starting materials, advantages and disadvantages. In particular, how about the utilization of catalysts. These issues should be carefully reviewed and discussed, and the present texts seemed too simple and lack of persuasiveness. Kindly suggest that the authors should let readers to believe the importance of the present work.
Response 1 : Thanks very much for your suggestion. Some classical synthetic routes of 2-TBM have been added in Introduction, and the advantages and disadvantages for different kinds of catalysts have been described in our manuscript.
Point 2: Secondly, as for the characterizations of organic compounds including catalysts, in addition to 1H NMR, FT-IR and TGA, 13C NMR, HR-MS, elemental analysis, melting point should be added. I have to clearly point out that only 1H NMR, FT-IR and TGA could not illustrate the structures of organic compounds including catalysts. This is very important, please also discuss with other experts on this point.
Response 2 : Thanks for your suggestion. The typical characteristic for deep eutectic solvent is easy to synthesis, high purity can be obtained just by mixing the hydrogen-bonding acceptor and hydrogen-bonding donor, the result of 1H NMR indicated the formation of hydrogen bonds in CAL-TsOH, and the result of FT-IR exhibited the characteristic functional groups in the catalyst, in addition, TGA indicated the good thermostability of CAL-TsOH, the characterizations of the DES in the manuscript were the common method in the literature. We also think the characterization for the catalyst by 13C NMR, HR-MS, elemental analysis, melting point will be beneficial, however, the time was really very limited, and the three characterizations for the catalyst could illustrate the structure of the catalyst.
Point 3: Next, very necessary, the authors will have to study the by-products of this kind of transformation, through GC-MS or HPLC-MS. Are there any by-products like esters or carbonyl compounds? Some key experimental evidences should be provided.
Response 3 : Thanks for your question. We have study the by-product of the reaction, there is very little butylated hydroxytoluene can be found by GC, and it can be negligent as the amount was too little.
Point 4: At last, the authors should summarize catalytic mechanism, which meant a lot to understanding the role of organo-catalyst in transformation.
Response 4 : The catalytic synthesis of 2-tert-butyl-4-methylphenol is a classical alkylation reaction, although a new catalytic system was established, and the catalytic mechanism was no difference from any other synthesis routes catalyzed by other acids. At the beginning, the carbonium ion intermediate of tertiary butyl produced catalyzed by CAL-TsOH, and then the carbonium ion intermediate of tertiary butyl reacted with p-cresol, and 2-tert-butyl-4-methylphenol was obtained.
Round 2
Reviewer 1 Report
Authors addressed major issues of the manuscript that can be published in this form on Catalysts.
Author Response
Dear reviewer, thank you very much for taking your time to review our manuscript submitted to Catalysts. We also want to express our appreciation for your valuable suggestions.
Reviewer 3 Report
Some important issues like C NMR and other characterizations were still not performed. I suggest the authors could improve.
Author Response
Point1: Some important issues like C NMR and other characterizations were still not performed. I suggest the authors could improve.
Response1: Dear reviewer, thank you very much for taking your time to review our manuscript submitted to Catalysts. We also want to express our appreciation for your valuable suggestions. The 13C NMR has been added in manuscript, and the HR-MS and DSC has been added in supporting information.